# ANNOTATION BY CLICKS: A POINT-SUPERVISED CONTRASTIVE VARIANCE METHOD FOR MEDICAL SEMANTIC SEGMENTATION

## ABSTRACT

Medical image segmentation methods typically rely on numerous dense annotated images for model training, which are notoriously expensive and time-consuming to collect. To alleviate this burden, weakly supervised techniques have been exploited to train segmentation models with less expensive annotations. In this paper, we propose a novel point-supervised contrastive variance method (PSCV) for medical image semantic segmentation, which only requires one pixel-point from each organ category to be annotated. The proposed method trains the base segmentation network by using a novel contrastive variance (CV) loss to exploit the unlabeled pixels and a partial cross-entropy loss on the labeled pixels. The CV loss function is designed to exploit the statistical spatial distribution properties of organs in medical images and their variance distribution map representations to enforce discriminative predictions over the unlabeled pixels. Experimental results on two standard medical image datasets demonstrate that the proposed method outperforms the state-of-the-art weakly supervised methods on point-supervised medical image semantic segmentation tasks.

## 1 INTRODUCTION

Medical image analysis has become indispensable in clinical diagnosis and complementary medicine with the developments in medical imaging technology. Notably, medical image segmentation is among the most fundamental and challenging tasks in medical image analysis, aiming to identify the category of each pixel of the input medical image. Existing methods have produced desirable results by using lots of annotated data to train convolutional neural networks. However, acquiring a large number of dense medical image annotations often requires specialist clinical knowledge and is time-consuming, which has bottlenecked the development of the existing field. To reduce the costly annotation requirement, weakly supervised learning technologies have been proposed to investigate the utilization of less expensive training labels for performing complex tasks.

Several types of weakly supervised signals have attracted many attentions, such as point-level Bearman et al. (2016); Qian et al. (2019); Tang et al. (2018), scribble-level Lin et al. (2016); Tang et al. (2018), box-level Dai et al. (2015); Khoreva et al. (2017), and image-level Ahn & Kwak (2018) supervision. As one of the simplest methods of annotating objects, point-level supervision can afford a trade-off between cost and information because they are significantly less expensive than dense pixel-level annotation yet contain important location information Bearman et al. (2016); Chen et al. (2021a). Therefore, we concentrate on segmenting organs in medical images under only the weak supervision provided by point annotations, in which each category of organs is annotated with just one pixel point, as shown in Figure 1 (a).

Some recent methods focus on using prior knowledge of the image, such as edge Qu et al. (2020); Yoo et al. (2019), size Kervadec et al. (2019), or spatial information Li et al. (2019), as auxiliary information to train the segmentation network or perform multi-stage refinement. However, their prior hypotheses may encounter difficulties when dealing with multi-class medical images, as similar appearances between different organs can produce noisy contexts to each other and hence inevitably interfere with the model learning. In addition, several approaches propose to mimic full supervision via generating pseudo-labels through dual structures Luo et al. (2022) and consistency constraints

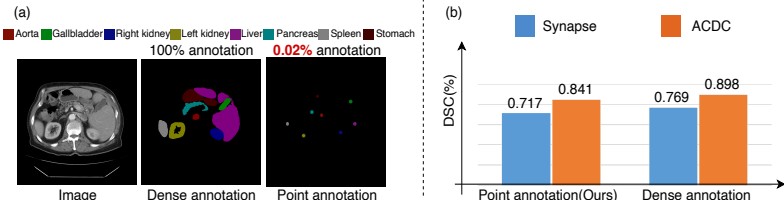

Figure 1: Annotation scenarios and performances of the proposed method. (a) Illustrations of dense and point annotations. (b) With only point annotation, the proposed method produces performances close to the ones produced with standard dense annotations based on UNet on two datasets.

Liu et al. (2022). But as the pseudo-labels are predicted from the weak model, the erroneous prediction problem can hardly be solved by directly using the predicted pseudo-labels as explicit training targets.

The overall fundamental challenges for point-supervised medical semantic segmentation lie in the following two aspects: (1) The appearance discrepancies between different organ regions of medical images are typically insignificant, and hence the organ boundaries are usually blurred, which can easily lead to over-segmentation problems. (2) Learning discriminatory information from only one pixel of annotation per organ is susceptible to underfitting. Inspired by the cognitive psychology fact that humans can correct and adjust visual information gradually by consciously identifying each region of interest Rensink (2000); Corbetta & Shulman (2002), it is desirable to investigate models to tackle the abovementioned challenges and enable point-supervised medical semantic segmentation in a similar manner.

In this paper, we propose a novel point-supervised contrastive variance (PSCV) method for learning medical image semantic segmentation models with only one pixel per class being labeled in each training image. The center of this approach is a novel contrastive variance loss function, which is designed to exploit unlabeled pixels to train the segmentation network together with a partial cross-entropy loss on the labeled pixels. This contrastive loss function is specifically designed for solving the point-supervised medical image segmentation task, which is computed over the pixel-level variance distribution map instead of the intermediate features. The standard Mumford-Shah function Mumford & Shah (1989) has been shown to be effective in image segmentation Vese & Chan (2002) by approximating the image using a smooth function inside each region. But its capacity for exploiting the unlabeled pixels is limited without considering the discrepancy information across different categories. The proposed contrastive variance loss function overcomes this drawback by capturing the statistical spatial distribution properties of the objects (i.e., organs) in medical images to eliminate the complex background of each category of organs in a contrastive manner. Moreover, by adopting a pixel-level variance distribution map representation for each organ in an image and contrasting the inter-image similarities between positive object pairs and negative object pairs, the proposed loss can help enforce compact predictions of each organ of the unlabeled pixels, and therefore eliminate the boundary prediction noise. The proposed PSCV is trained in an end-to-end manner. It largely reduces the performance gap between point-annotated segmentation methods and the fully supervised method, as shown in Figure 1 (b). In summary, this work makes the following contributions:

- We propose a novel PSCV method to tackle the medical image semantic segmentation problem by exploiting both unlabeled and labeled pixels with limited point supervisions: one labeled pixel point per class in each image.

- We design a novel contrastive variance function that exploits the spatial distribution properties of the organs in medical images and their variance distribution map representations to enforce discriminative predictions over the unlabeled pixels.

- Experimental results on two medical image segmentation datasets show that the proposed PSCV can substantially outperform the state-of-the-art weakly supervised medical segmentation methods and greatly narrow the gap with the results of the fully supervised method.

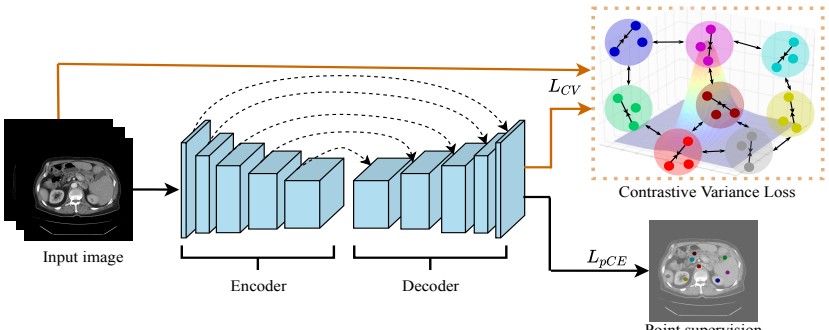

Figure 2: An overview of the proposed method. In the training stage, we first input the images into the segmentation network and obtain predictions through the encoder and decoder. Then we calculate two loss functions, a partial cross entropy loss $L_{pCE}$ and a contrastive variance Loss $L_{CV}$, to update the segmentation network. The $L_{pCE}$ is calculated using point annotations and predictions, while the $L_{CV}$ is calculated using the input images and predictions. In the testing stage, we input the test image into the segmentation network to obtain the segmentation result.

## 2 RELATED WORK

### 2.1 WEAK SUPERVISION FOR IMAGE SEGMENTATION

Weakly-supervised segmentation methods have proposed to use different types of weak supervision information, such as point-level Bearman et al. (2016); Qian et al. (2019); Tang et al. (2018), scribble-level Lin et al. (2016); Tang et al. (2018), box-level Dai et al. (2015); Khoreva et al. (2017), and image-level Ahn & Kwak (2018) supervisions. In particular, Lin et al. Lin et al. (2016) proposed to propagate information from scribble regions to unlabeled regions by generating proposals. Tang et al. Tang et al. (2018) combined the partial cross entropy for labeled pixels and the normalized cut to unlabeled pixels to obtain a more generalised model. Kim and Ye Kim & Ye (2019) proposed a novel loss function based on Mumford-Shah functional that can alleviate the burden of manual annotation. Qian et al. Qian et al. (2019) considered both the intra- and inter-category relations among different images for distance metric learning. In order to obtain reasonable pseudo-labels, some methods stack temporal level predictions Lee & Jeong (2020) or constrain the consistency of different outputs from the model to increase the performance Ouali et al. (2020); Chen et al. (2021c). The performances of the above methods nevertheless still have considerable gaps from fully supervised methods and their study has focused on natural scene images.

### 2.2 MEDICAL IMAGE SEGMENTATION WITH WEAK SUPERVISION

Medical image semantic segmentation with weak supervision poses different challenges from the segmentation task for natural scene images Luo et al. (2022); Dorent et al. (2021); Roth et al. (2021); Qu et al. (2020); Tajbakhsh et al. (2020). Some works have explored using extreme points as supervisory signals for medical image segmentation Dorent et al. (2021); Roth et al. (2021). Roth et al. Roth et al. (2021) used the Random Walker according to extreme points to generate initial pseudo-labels, which were then used to train the segmentation model. Subsequently, Dorent et al. Dorent et al. (2021) generated more annotated voxels by deep geodesics connecting and used a CRF regularised loss to obtain better performance. Kervadec et al. Kervadec et al. (2019) proposed a loss function that constrains the prediction by considering prior knowledge of size. Laradji et al. Laradji et al. (2021) constrained the output predictions to be consistent through geometric transformations. Can et al. Can et al. (2018) introduced an iterative two-step framework that uses conditional random field (CRF) to relabel the training set and then trains the model recursively. Recently, Luo et al. Luo et al. (2022) proposed a dual-branch network to generate pseudo-labels dynamically, which learns from scribble annotations in an end-to-end way. Although the above work has made some progress, they have not yet satisfactorily addressed the problem of weakly supervised semantic segmentation of medical organs, and more research effort is needed on this task.

## 3 PROPOSED METHOD

In this section, we present a novel point-supervised contrastive variance method for medical semantic segmentation (PSCV), which exploits both labeled and unlabeled pixels to learn discriminative segmentation models by capturing the statistical spatial distribution properties of organs in medical images in a contrastive manner.

In particular, we assume weakly point-annotated data provided by only making a few clicks on each training image. That is, in each training image, only one pixel from each organ category is annotated. We aim to train a good segmentation model from $N$ point-annotated training images, $\mathcal{D} = \{(I^n, Y^n)\}_{n=1}^N$, where $I^n \in \mathbb{R}^{1*H*W}$ denotes the $n$-th training image, $Y^n \in \{0, 1\}^{K*H*W}$ denotes the corresponding point-supervised label matrix, $K$ is the number of classes (i.e., organ categories) in the dataset, $H$ and $W$ are the height and width of the input image, respectively. Let $\Omega \subset \mathbb{R}^2$ denote the set of spatial locations for the input image. Since each point-annotated training image $I^n$ only has $K$ pixels being labeled, the subset of spatial locations for these labeled pixels can be denoted as $\Omega_y^n \subset \Omega$, and the point-supervised pixel-label pairs in the $n$-th image can be written as $\{(I^n(r), Y^n(r)), \forall r \in \Omega_y^n\}$.

The pipeline of the proposed method is illustrated in Figure 2. Each image first goes through a base segmentation network to produce prediction outputs. The overall training loss is then computed as two parts: the standard partial cross-entropy loss $L_{pCE}$ computed based on the given point annotations and the corresponding prediction outputs, and the proposed contrastive variance Loss $L_{CV}$ computed based on the whole input image and all the prediction outputs. The segmentation network consists of an encoder $f_{encoder} : \mathbb{R}^{1*H*W} \rightarrow \mathbb{R}^{c*h*w}$ and a decoder $f_{decoder} : \mathbb{R}^{c*h*w} \rightarrow \mathbb{R}^{K*H*W}$, where $c, h,$ and $w$ are the number of channels, height, and width of the intermediate embedding features respectively. The proposed method is trained end-to-end with the joint training loss. Below we elaborate the proposed method by first introducing the base point-supervised segmentation training, then revisiting the standard Mumford-Shah loss function for image segmentation, and finally presenting the proposed novel contrastive variance loss function, as well as the training and test procedures of the proposed method.

### 3.1 POINT-SUPERVISED LEARNING

Following the framework in Figure 2, the prediction output $\hat{Y}^n \in [0, 1]^{K*H*W}$ for a training image $I^n$ can be generated through the pipeline of an encoder $f_{encoder}$ and a decoder $f_{decoder}$, as follows:

$$\hat{Y}^n = \mathrm{softmax}(f_{decoder} \circ f_{encoder}(I^n)), \tag{1}$$

where $\mathrm{softmax}(\cdot)$ denotes the class-wise softmax function that predicts each pixel into a probability vector of $K$ classes. Let $\hat{Y}_k^n(r)$ denote the predicted probability of a pixel at location $r$ belonging to the $k$-th class. We then have $\sum_{k=1}^K \hat{Y}_k^n(r) = 1$.

In general point-supervised learning, especially given a fraction of labeled points, the partial cross-entropy loss $L_{pCE}$ is typically minimized to train the segmentation network Tang et al. (2018). Here given the few annotated points indicated by $\Omega_y^n$ for each training image $I^n$, the cross-entropy loss $L_{pCE}$ is computed by using the point-supervised ground truth $Y^n$ and the prediction output/mask $\hat{Y}^n$ as follows:

$$L_{pCE} = -\sum_{n=1}^N \sum_{r \in \Omega_y^n} \sum_{k=1}^K Y_k^n(r) \log \hat{Y}_k^n(r) \tag{2}$$

### 3.2 REVISITING MUMFORD-SHAH LOSS FUNCTION

In this subsection, we provide an overview of the Mumford-Shah loss function Kim & Ye (2019), which treats image segmentation as a Mumford-Shah energy minimization problem Mumford & Shah (1989). It aims to segment images into $K$ classes of piece-wise constant regions by enforcing each region to have similar pixel values with the regularization of the contour length.

Specifically, by using the prediction function output $\hat{Y}_k$ as the characteristic function for the $k$-th class Kim & Ye (2019), the Mumford-Shah loss functional $L_{MS}$ is defined as below:

$$L_{MS} = \lambda_{ms} \sum_{n=1}^{N} \sum_{k=1}^{K} \int_{\Omega} |I^n(r) - c_k^n|^2 \hat{Y}_k^n(r) dr + \mu \sum_{n=1}^{N} \sum_{k=1}^{K} \int_{\Omega} |\nabla \hat{Y}_k^n(r)| dr, \qquad (3)$$

where $c_k^n \in \mathbb{R}^{1*1*1}$ is the average pixel value of the $k$-th class in the $n$-th image; the gradient $\nabla \hat{Y}_k^n(r) = \frac{\partial \hat{Y}_k^n(r)}{\partial r}$ computes the horizontal and vertical changes in the predictions, and is used in the second term to enforce the predictions to be smooth within each of the organs.

### 3.3 Contrastive Variance Loss Function

It is very challenging to identify boundaries between different classes with just a single point annotation per class in an image. The partial cross-entropy loss on the labeled pixels can only provide limited guidance. It is therefore essential to exploit the unannotated pixels. Although the Mumford-Shah loss above exploits the unlabeled pixels in a predictive K-means clustering manner, it still lacks sufficient discriminative capacity for separating different categories and identifying the segmentation boundaries. Meanwhile, medical images have some unique properties for the spatial distributions of different categories of organs—even with operations such as random rotation and flipping, the spatial locations of the same organ, especially the smaller ones, in different images will still be within some limited regions in contrast to the whole image background. Inspired by this observation, we develop a novel contrastive variance loss function to exploit the statistical spatial category distribution information in a contrastive manner, aiming to enhance the discrepancy between each organ and its background.

Specifically, we first calculate the pixel-level variance distribution map $z_k^n \in \mathbb{R}^{1*H*W}$ for each $k$-th organ category on the $n$-th image, such as:

$$z_k^n(r) = |I^n(r) - c_k^n|^2 \hat{Y}_k^n(r), \forall r \in \Omega, \qquad (4)$$

where $c_k^n \in \mathbb{R}^{1*1*1}$ is the mean pixel value for the $k$-th category region on the $n$-th image, which is defined as:

$$c_k^n = \frac{\int_{\Omega} I^n(r) \hat{Y}_k^n(r) dr}{\int_{\Omega} \hat{Y}_k^n(r) dr}. \qquad (5)$$

We then use this variance distribution map $z_k^n$ as the appearance representation of the $k$-th class in the $n$-th image and define the following contrastive variance loss to promote the inter-image similarity between the same class of organs in contrast to the similarities between different classes:

$$L_{CV} = \lambda_{cv} \sum_{n=1}^{N} \sum_{k=1}^{K} - \log \frac{pos}{pos + neg} + \mu \sum_{n=1}^{N} \sum_{k=1}^{K} \int_{\Omega} |\nabla \hat{Y}_k^n(r)| dr, \qquad (6)$$

with

$$pos = \exp(\cos(z_k^n, z_k^{m_n})/\tau), neg = \sum_{i \neq n} \sum_{j \neq k} \exp(\cos(z_k^n, z_j^i)/\tau), \qquad (7)$$

where $\cos(\cdot, \cdot)$ denotes the cosine similarity function, and $\tau$ is the temperature hyper-parameter. In this loss function, given the pixel variance distribution map $z_k^n$ for the $k$-th category on the $n$-th training image, we build the contrastive relationship by randomly selecting another $m_n$-th image that contains the same $k$-th category and using $z_k^{m_n} \in \{z_k^i, i \neq n | z_k^n\}$ as a positive sample for $z_k^n$, while using all pixel variance distribution maps for all the other categories $\{j : j \neq k\}$ on all other training images $\{i : i \neq n\}$ as negative samples for $z_k^n$.

Different from the typical contrastive losses popularly deployed in the literature, which are computed over intermediate features Khosla et al. (2020); Hu et al. (2021), our proposed contrastive variance loss above is specifically designed for the point-supervised medical image segmentation task and is computed over the pixel-level variance distribution map. This contrastive variance (CV) loss integrates both the variance and contrastive learning paradigms in a way that has not been attempted, and is designed to possess the following advantages. First, under this CV loss, by maximizing the similarities between the positive pairs in contrast to the negative pairs, we can enforce each prediction $\hat{Y}_k^n$ into the statistically possible spatial regions for the $k$-th category across all the training images.

Table 1: Quantitative comparison results on the Synapse dataset. We report the class average DSC and HD95 results and the DSC results for all individual classes. The best results are in bold-font and the second best are underlined.

| Method | HD95.Avg↓ | DSC.Avg↑ | Aor | Gal | Kid(L) | Kid(R) | Liv | Pan | Spl | Sto |
|---|---|---|---|---|---|---|---|---|---|---|
| pCE Lin et al. (2016) | 112.83 | 0.469 | 0.348 | 0.251 | 0.522 | 0.443 | 0.792 | 0.257 | 0.713 | 0.426 |
| EntMini Grandvalet & Bengio (2004) | 105.19 | 0.485 | 0.329 | 0.257 | 0.577 | 0.491 | 0.787 | 0.304 | 0.663 | 0.471 |
| WSL4MIS Luo et al. (2022) | 81.79 | 0.497 | 0.315 | 0.189 | 0.625 | 0.526 | 0.817 | 0.290 | 0.722 | 0.491 |
| USTM Liu et al. (2022) | 97.05 | 0.552 | 0.443 | 0.312 | 0.645 | 0.631 | 0.834 | 0.311 | 0.761 | 0.482 |
| GatedCRF Obukhov et al. (2019) | 66.32 | 0.596 | 0.457 | 0.420 | 0.722 | 0.666 | 0.836 | 0.361 | 0.730 | 0.578 |
| PSCV (Ours) | 37.38 | 0.717 | 0.803 | 0.572 | 0.770 | 0.736 | 0.901 | 0.522 | 0.840 | 0.587 |
| FullySup | 39.70 | 0.769 | 0.891 | 0.697 | 0.778 | 0.686 | 0.934 | 0.540 | 0.867 | 0.756 |

Table 2: Quantitative comparison results on the ACDC Dataset. We report the class average results and the results for all individual classes n terms of the DSC and HD95 metrics. The best results are in bold-font and the second best are underlined.

| Method | DSC.Avg↑ | RV | Myo | LV | HD95.Avg↓ | RV | Myo | LV |
|---|---|---|---|---|---|---|---|---|
| pCE Lin et al. (2016) | 0.722 | 0.622 | 0.684 | 0.860 | 39.30 | 40.22 | 40.60 | 37.08 |
| EntMini Grandvalet & Bengio (2004) | 0.740 | 0.632 | 0.718 | 0.870 | 28.56 | 25.79 | 30.76 | 29.14 |
| USTM Liu et al. (2022) | 0.767 | 0.691 | 0.736 | 0.874 | 18.35 | 17.63 | 15.33 | 22.08 |
| WSL4MIS Luo et al. (2022) | 0.768 | 0.664 | 0.745 | 0.896 | 10.00 | 10.73 | 9.37 | 9.92 |
| GatedCRF Obukhov et al. (2019) | 0.814 | 0.743 | 0.788 | 0.911 | 4.03 | 7.45 | 2.47 | 2.17 |
| PSCV (Ours) | 0.841 | 0.810 | 0.804 | 0.910 | 3.57 | 3.42 | 2.98 | 4.33 |
| FullySup | 0.898 | 0.882 | 0.883 | 0.930 | 7.00 | 6.90 | 5.90 | 8.10 |

This can help eliminate the complex background formed by all the other categories and improve the discriminative capacity of the segmentation model. Second, by using the pixel-level variance distribution maps as the appearance representation of the organs for similarity computation, one can effectively eliminate the irrelevant inter-image pixel variations caused by different collection equipment or conditions. Third, using the cosine similarity function to compute the similarity between a pair of $(z_k^n, z_k^{m_n})$ is equivalent to computing the dot product similarity between two normalized probability distribution vectors. Hence, the smaller effective length of the vectors will lead to larger similarities. This property consequently will push each organ prediction $\hat{Y}_k^n$ to cover more compact regions for more accurate predictions of the unlabeled pixels, while eliminating the boundary noise. Overall, we expect this novel loss function can make effective use of all the unlabeled pixels to support few-point-annotated segmentation model training.

### 3.4 TRAINING AND TESTING PROCEDURES

The proposed PSCV is straightforward and can be trained in an end-to-end manner. It does not require any additional layers or learnable parameters on top of the base segmentation network. During the training phase, the learnable parameters of the segmentation network are learned to minimize the following joint loss function:

$$L_{total} = L_{pCE} + L_{CV}, \tag{8}$$

where $L_{pCE}$ encodes the supervised information in the labeled pixels and $L_{CV}$ exploits all pixels in the training images in a self-supervised manner. Note for batch-wise training procedures, these losses are computed within each batch. During testing, each test image $I$ can be given as an input to the trained segmentation network (encoder and decoder) to obtain the prediction results $\hat{Y}$.

## 4 EXPERIMENTAL RESULTS

### 4.1 DATASETS AND EVALUATION METRICS.

Following Wang et al. (2022); Chen et al. (2021b), we benchmark the proposed PSCV on two medical image datasets: the Synapse multi-organ CT dataset and the Automated cardiac diagnosis

Table 3: Ablation study over the proposed method on Synapse. We report the class average DSC and HD95 results and the DSC results for all individual classes. pCE: using only the $L_{pCE}$ loss. vanilla MS: using the combined loss: $L_{pCE} + L_{MS}$.

| Method | HD95.Avg↓ | DSC.Avg↑ | Aor | Gal | Kid(L) | Kid(R) | Liv | Pan | Spl | Sto |
|---|---|---|---|---|---|---|---|---|---|---|
| pCE | 112.83 | 0.469 | 0.348 | 0.251 | 0.522 | 0.443 | 0.792 | 0.257 | 0.713 | 0.426 |
| vanilla MS | 41.58 | 0.634 | 0.596 | 0.460 | 0.717 | 0.678 | 0.892 | 0.357 | 0.781 | **0.588** |
| PSCV | **37.38** | **0.717** | **0.803** | **0.572** | **0.770** | **0.736** | **0.901** | **0.522** | **0.840** | 0.587 |

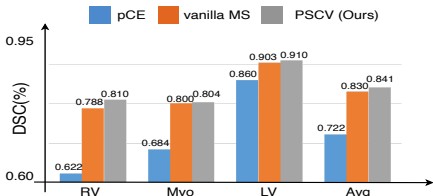

Figure 3: Ablation study of the proposed PSCV on the ACDC dataset. The horizontal coordinates indicate the different class categories and the average. The vertical coordinates indicate the DSC values.

challenge (ACDC) dataset. The Synapse dataset consists of 30 abdominal clinical CT cases with 2211 images and contains eight organ categories. Following Wang et al. (2022); Chen et al. (2021b), 18 cases are used for training and 12 cases are used for testing. The ACDC dataset contains 100 cardiac MRI examples from MRI scanners with 1300 images and includes three organ categories: left ventricle (LV), right ventricle (RV), and myocardium (Myo). We evaluate the proposed PSCV on the ACDC via five-fold cross-validation Luo et al. (2022). Following previous works Ronneberger et al. (2015); Wang et al. (2022); Chen et al. (2021b), the Dice Similarity Coefficient (DSC) and the 95% Hausdorff Distance (HD95) are used as evaluation metrics.

## 4.2 IMPLEMENTATION DETAILS

We randomly select one pixel from the ground truth mask of each category as labeled data to generate point annotations for each training image. The most widely used medical image segmentation network UNet Ronneberger et al. (2015) is used as the base architecture. In the training stage, we first normalized the pixel values of input images to [0,1]. Then the input images are resized to 256×256 with random flipping and random rotation Luo et al. (2022). The weights of the proposed model are randomly initialized. The model is trained by stochastic gradient descent (SGD) with a momentum of 0.9, a weight decay of 1e-5, a batch size of 12, and a poly learning rate policy with a power of 0.9. We set the hyperparameters $\mu$=1e-5 and $\tau$=0.07. We use a learning rate of 0.01 for 60K total iterations and $\lambda_{cv}$=1e-3 on the ACDC dataset and a learning rate of 0.001 for 100K total iterations and $\lambda_{cv}$=3e-1 on the Synapse dataset. Inspired by the central bias theory Levy et al. (2001), in the testing stage we filtered fifty pixels that are close to the left and right edges of the prediction as the background to further refine the results on Synapse for all the methods.

## 4.3 QUANTITATIVE EVALUATION RESULTS

The proposed PSCV is compared with several state-of-the-art methods on the Synapse and the ACDC datasets: pCE Lin et al. (2016) (lower bound), EntMini Grandvalet & Bengio (2004), WSL4MIS Luo et al. (2022), USTM Liu et al. (2022) and GatedCRF Obukhov et al. (2019). For fair comparisons, we use UNet as the backbone for all the comparison methods, while all of them are trained with the same point-annotated data. In addition, we also tested the fully supervised model trained with the cross-entropy loss, which uses the dense annotations of all pixels in the training images, and its performance can be treated as a soft upper bound for the point-supervised methods.

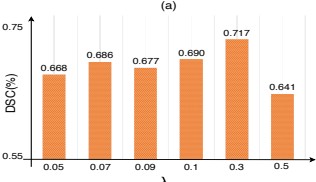 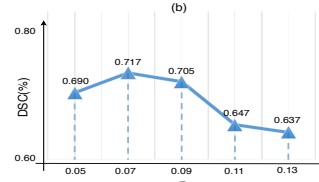

Figure 4: (a) Impact of the weight of the proposed contrastive variance loss, $\lambda_{cv}$, on the Synpase dataset. (b) Impact of the hyper-parameter $\tau$ of the proposed contrastive variance loss on the Synapse dataset. The horizontal coordinate indicates the $\tau$ value and the vertical coordinate indicates the class average DSC value.

### 4.3.1 COMPARISON RESULTS ON SYNAPSE

The test results for all the comparison methods on the Synapse dataset are reported in Table 1. We can see that the proposed PSCV performs substantially better than the other comparison methods in terms of both DSC and HD95 metrics. Note that WSL4MIS and USTM are based on pseudo-label generations. Due to the extremely small number of annotated pixels, the noise of their generated pseudo-labels tends to bias the model learning. The proposed PSCV outperforms these two approaches by 22% and 16.5% respectively, and achieves 0.717 in terms of the class average of DSC results. Moreover, PSCV outperforms the second best result of GatedCRF by 12.1% in terms of the class average DSC, and substantially reduces the class average HD95 result from 66.32 to 37.38. Aorta (Aor) and Liver (Liv) are the smallest and largest organs in the Synapse dataset, while the proposed PSCV achieves very satisfactory results (0.803 and 0.901 in terms of class average DSC) in both categories compared to the other methods. It is also worth noting that the performance of PSCV is very close to the upper bound produced by the fully supervised method. These results validate the effectiveness of the proposed PSCV.

### 4.3.2 COMPARISON RESULTS ON ACDC

The comparison results on the ACDC dataset are reported in Table 2. It is clear that the proposed PSCV outperforms the state-of-the-art comparison methods, achieving the best class average DSC value of 0.841 and the best class average HD95 value of 3.57. In particular, PSCV outperforms the second best, GatedCRF, by 2.7% in terms of class average DSC. PSCV also substantially reduces the performance gap between the point-supervised methods and the fully supervised method in terms of DSC. In terms of HD95, PSCV even outperforms the fully supervised method. These results suggest that PSCV is effective for point-supervised learning.

### 4.4 ABLATION STUDY

To demonstrate the effectiveness of the proposed CV loss, we conducted an ablation study by comparing the full PSCV method with two variant methods: (1) "pCE" denotes the variant that performs training only using the partial cross-entropy loss $L_{pCE}$; (2) "vanilla MS" denotes the variant that replaces the CV loss, $L_{CV}$, in the full PSCV method with the standard Mumford-Shah loss, $L_{MS}$. The comparison results on the Synapse and the ACDC datasets are reported in Table 3 and Figure 3 respectively. From Table 3, we can see that using $L_{pCE}$ alone produces poor segmentation results. With the additional $L_{MS}$ loss, "vanilla MS" improves the segmentation results by 16.5% in terms of class-average DSC and by 71.25 in terms of class-average HD95. By using the proposed contrastive variance loss $L_{CV}$, PSCV further improves the performance by another 8.3% in terms of DSC and by 4.2 in terms of HD95. Similar observations can be made from the results reported in Figure 3 as well. Overall, the proposed CV loss is more effective than the regular Mumford-Shah loss.

### 4.5 HYPERPARAMETER ANALYSIS

### 4.5.1 IMPACT OF THE WEIGHT OF THE CV LOSS FUNCTION

We summarize the experimental results regarding the influence of the weight of the CV loss function, $\lambda_{cv}$, in Eq.(6) on the Synapse dataset in Figure 4 (a). The experiments are conducted by fixing the

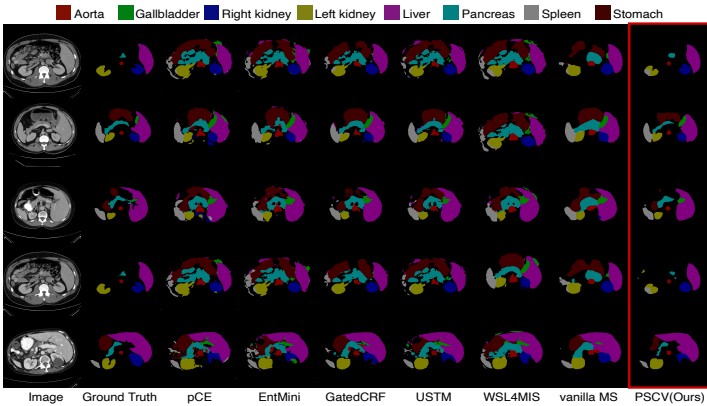

Figure 5: Visualized examples of the segmentation results for the proposed PSCV and other state-of-the-art methods on Synapse. The first and the second columns show the input images and the ground-truth. The last column shows visualized examples of the segmentation results produced by PSCV. The other columns present the results obtained by other methods.

other hyperparameters and only adjusting the value of $\lambda_{cv}$. Combining the results in Figure 4 (a) and Table 3, we can see that for a range of different $\lambda_{cv}$ values within $[0.05, 0.3]$, the proposed CV loss can outperform the Mumford-Shah loss, while the best performance is produced when $\lambda_{cv} = 0.3$. However, when $\lambda_{cv}$ gets too large, e.g., $\lambda_{cv} = 0.5$, the performance degrades substantially. This suggests that though the unsupervised loss term is useful, it still should be an auxiliary loss and should not dominate the supervised partial cross-entropy loss.

### 4.5.2 IMPACT OF THE TEMPERATURE OF THE CV LOSS

We illustrate the impact of the temperature $\tau$ of the contrastive variance Loss in Eq.(6) on the segmentation performance in Figure 4 (b). It demonstrates that the temperature $\tau$ is crucial for controlling the strength of penalties on hard negative samples in the contrast learning paradigm Wang & Liu (2021). As shown in Figure 4 (b), good experimental results can achieved with smaller $\tau$ values within $[0.05, 0.09]$ and the best result is produced with $\tau = 0.07$. With $\tau$ values larger than 0.09, the performance of the proposed approach degrades substantially. This experimental phenomenon suggests using smaller $\tau$ values.

### 4.6 QUALITATIVE EVALUATION RESULTS

We present a visualized comparison between our results and the results produced by several state-of-the-art methods in Figure 5. We can see that the various organs differ significantly in size and shape in medical organ images, and the background regions are frequently mistaken for organs by many methods. Additionally, different image samples have different intensities of the same organs. From Figure 5, we can see that benefiting from using the contrastive paradigm and the variance distribution map representations in designing the CV loss, the proposed PSCV can better discriminate the organs from the background, and produce more compact and accurate segmentations than the other comparison methods. Its ability to accurately segment the smaller organs is especially notable.

## 5 CONCLUSIONS

In this paper, we proposed a novel method, PSCV, for point-supervised medical image semantic segmentation. PSCV adopts the partial cross-entropy loss for the labeled point pixels and a novel contrastive variance loss for the unlabeled pixels. The proposed contrastive variance loss exploits the spatial distribution properties of the medical organs and their variance distribution map representations in a contrastive paradigm to enforce the model's discriminative and smooth segmentation capacity. Experimental results demonstrate the proposed PSCV substantially outperforms existing state-of-the-art methods on two medical image datasets, and its ability to accurately segment the more difficult smaller organs are particularly notable.

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
