# Supplementary materials of Annotation by Clicks: A Point-Supervised Contrastive Variance Method for Medical Semantic Segmentation

## Abstract

In this section, we demonstrate more experimental results, including the comparison with the state-of-the-art methods; the impact of the proposed contrastive variance loss; the impact of the weight of the proposed contrastive variance loss; the impact of the temperature of the contrastive variance loss; the impact of using combined points as pseudo labels; the impact of contrasting variance maps; the impact of sampling different points; the visualization of $z_k^n$.

## 1 Additional Quantitative Evaluation Results

**Comparison with the State-of-the-Art Methods.** We summarize the comparison of the state-of-the-art methods on Synapse in terms of the class average HD95 results and the HD95 results for all individual classes in Table 3. As shown in this table, the proposed method outperforms all the state-of-the-art methods in terms of HD95, with the mean value of HD95 of 37.38. The proposed PSCV achieves the best results for most categories, especially for Aor, Gal and Pan categories. It is worth noting that the second and third best methods are GatedCRF and USTM. The former proposes regularized loss with a gate mechanism, while the latter is concerned with generating better pseudo-labels. For the Aor and Gal categories, the proposed method achieves the best results with the HD95 value of 15.33 and 39.26, respectively, and exceeds the second best results with 44.59 and 69.13. In addition, the Aorta and Liver are the smallest and largest organs in the dataset in the Synapse dataset. Surprisingly, the proposed method works better than other methods in both categories.

To further demonstrate the effectiveness of our proposed PSCV, we also compared some related methods (including Tang et al. (2018); Zhao & Yin (2020); Zhai et al. (2023); He et al. (2022)) in Table 2. We reproduced these methods based on their codes or methods described in the paper and applied them to our proposed experimental setup. Tang et al. (2018) proposed a regularized loss for weakly supervised segmentation. Zhao & Yin (2020) addresses self-deception and unstable training through hybrid-training schemes incorporating divergence and consistency losses. Zhai et al. (2023) employs geodesic distance transform to expand seed points, introducing a multi-view CRF loss and Variance Minimization loss for segmentation model training. He et al. (2022) introduces intra-slice and inter-slice contrastive learning methods. Experimental results show that the proposed PSCV outperforms all the methods compared above on the Synapse dataset.

Table 1: Ablation study of the impact of the proposed contrastive variance loss on the ACDC dataset. We report the class average HD95 results and the results for all individual classes.

| Method | HD95.Avg↓ | RV | Myo | LV |
|---|---|---|---|---|
| pCE | 39.30 | 40.22 | 40.60 | 37.08 |
| vanilla MS | 4.12 | 4.81 | **2.36** | 5.20 |
| PSCV (Ours) | **3.58** | **3.42** | 2.98 | **4.33** |

Table 2: Quantitative comparison results on the Synapse dataset. We report the class average DSC.

| Method | PSCV (Ours) | Tang et al. (2018) | Zhao & Yin (2020) | Zhai et al. (2023) | He et al. (2022) |
|--------|-------------|--------------------|--------------------|--------------------|------------------|
| DSC | 0.717 | 0.568 | 0.590 | 0.520 | 0.553 |

## 2 ADDITIONAL ABLATION STUDIES

**Impact of the proposed loss function.** To demonstrate the impact of the proposed method, we summarize the results of various studies conducted on the ACDC dataset in terms of class average and the results for all individual classes of HD95 in Table 1. When using only pCE as a supervised loss, this method can achieve limited results, reaching the average HD95 value of 39.3. Compared to the "pCE" method, using "vanilla MS" can bring a significant improvement and achieves 4.12 in terms of the average HD95, which uses $L_{MS}$ as an auxiliary loss to help train the segmentation model. Furthermore, the proposed PSCV can achieve the best results on this dataset, reaching an HD95 mean of 3.58.

**Impact of using combined points as pseudo labels.** Based on the fact that the approximate locations of the organs in the medical images are all basically similar, we illustrate impact of using combined points as pseudo labels in Figure 1 (a). In this section, we use each point annotation in the dataset to compose a pseudo-label map and use it as a auxiliary training target to train the model. The experimental results show that simply merging all the point information to generate pseudo-labels is not enough to help the neural network obtain the desired results. In contrast, the proposed PSCV is able to obtain more accurate segmentation results.

**Impact of contrasting variance maps.** To demonstrate the impact of contrasting variance maps, we summarize the results of contrasting variance maps and contrasting intermediate features of UNet conducted on both the Synapse and the ACDC dataset in terms of DSC values in Figure 1 (b). Feature maps have lower resolutions and therefore many low-level characteristics are lost. When training the segmentation network, the pixels around the boundary will be centered on similar receptive fields. Consequently, the network might yield masks with less precise shapes and boundaries. Besides, the network's training always assumes pixel independence. Nonetheless, treating pixels as independent entities might not be optimal, as this disregards the potential correlations among them that could be harnessed for improved performance. In contrast, our contrastive variance loss function is computed over the pixel-level variance distribution map instead of the intermediate features. This unique approach enables our loss function to leverage variance information of pixel-level corresponding to different classes, thus effectively capturing the intricate pixel-level relationships and detailed image characteristics. Experimental results show that using our contrastive variance loss yields better performance than directly contrasting feature maps on the Synapse dataset in terms of DSC value

**Impact of sampling different points.** We present the impact of sampling different points on the Synapse dataset in terms of DSC in Figure 2. Five random samples of point labels were taken to demonstrate the results. The experimental results indicate that the proposed PSCV is robust, as randomly selecting different points as supervisory information has little effect on the results.

Table 3: Quantitative comparison of the medical semantic segmentation task on the Synapse dataset. We report the class average HD95 results and the HD95 results for all individual classes.

| Method | HD95.Avg↓ | Aor | Gal | Kid(L) | Kid(R) | Liv | Pan | Spl | Sto |
|--------|-----------|-----|-----|--------|--------|-----|-----|-----|-----|
| pCE | 112.83 | 69.84 | 170.07 | 125.96 | 139.00 | 114.97 | 74.49 | 119.27 | 89.04 |
| EntMini | 105.19 | 60.14 | 169.75 | 119.22 | 137.01 | 95.98 | 61.87 | 119.54 | 78.06 |
| WSL4 | 81.79 | 59.92 | 119.92 | 85.10 | 127.86 | 81.99 | 42.31 | 71.06 | 66.18 |
| USTM | 97.05 | 61.17 | 129.03 | 119.41 | 131.34 | 82.08 | 50.15 | 133.57 | 69.71 |
| GatedCRF | 66.32 | 61.02 | 108.39 | **30.90** | 126.04 | 47.75 | 36.78 | 86.43 | 33.27 |
| PSCV (Ours) | **37.38** | **15.33** | **39.26** | 53.53 | **80.30** | **28.09** | **19.61** | **40.06** | **22.88** |

Table 4: Ablation study of the weight of the proposed contrastive variance loss on the Synpase dataset. We report the class average DSC results and the results for all individual classes.

| $\lambda_{cv}$ | DSC.Avg↑ | Aor | Gal | Kid(L) | Kid(R) | Liv | Pan | Spl | Sto |
|---|---|---|---|---|---|---|---|---|---|
| 0.05 | 0.668 | 0.620 | 0.576 | 0.743 | 0.698 | 0.904 | 0.377 | 0.823 | 0.602 |
| 0.07 | 0.686 | 0.691 | 0.574 | 0.715 | 0.701 | 0.905 | 0.430 | 0.829 | **0.642** |
| 0.09 | 0.677 | 0.683 | 0.557 | 0.726 | 0.696 | **0.906** | 0.425 | 0.809 | 0.617 |
| 0.1 | 0.690 | 0.726 | **0.583** | 0.744 | 0.722 | **0.906** | 0.434 | 0.810 | 0.591 |
| 0.3 | **0.717** | **0.803** | 0.572 | **0.770** | 0.736 | 0.901 | **0.522** | 0.840 | 0.587 |
| 0.5 | 0.627 | 0.180 | 0.516 | 0.761 | **0.748** | 0.893 | 0.483 | **0.847** | 0.591 |

Table 5: Ablation study of the weight of the proposed contrastive variance loss on the Synpase dataset. We report the class average HD95 results and the results for all individual classes.

| $\lambda_{cv}$ | HD95.Avg↓ | Aor | Gal | Kid(L) | Kid(R) | Liv | Pan | Spl | Sto |
|---|---|---|---|---|---|---|---|---|---|
| 0.05 | 38.76 | 31.26 | 36.47 | **49.00** | 82.80 | 19.66 | 23.09 | 42.75 | 25.06 |
| 0.07 | **37.30** | 25.57 | **36.38** | 51.56 | 62.08 | 20.20 | 21.42 | 54.09 | 27.12 |
| 0.09 | 108.72 | 274.97 | 53.82 | 244.53 | 87.52 | 26.00 | 94.16 | 63.58 | 25.17 |
| 0.1 | 68.13 | 22.54 | 40.65 | 135.26 | 177.41 | **16.01** | 66.64 | 58.33 | 28.24 |
| 0.3 | 37.38 | **15.33** | 39.26 | 53.53 | 80.30 | 28.09 | 19.61 | **40.06** | **22.88** |
| 0.5 | 42.28 | 50.10 | 51.67 | 61.66 | **49.44** | 30.78 | **17.39** | 49.67 | 27.52 |

Table 6: Ablation study of the hyper-parameter $\tau$ of the proposed contrastive variance loss on the Synpase dataset. We report the class average DSC results and the results for all individual classes.

| $\tau$ | DSC.Avg↑ | Aor | Gal | Kid(L) | Kid(R) | Liv | Pan | Spl | Sto |
|---|---|---|---|---|---|---|---|---|---|
| 0.05 | 0.690 | 0.743 | 0.529 | 0.743 | 0.717 | 0.899 | 0.461 | 0.840 | 0.591 |
| 0.07 | **0.717** | **0.803** | **0.572** | 0.770 | 0.737 | 0.901 | **0.522** | 0.840 | 0.587 |
| 0.09 | 0.705 | 0.749 | 0.566 | 0.763 | 0.727 | **0.911** | 0.497 | **0.844** | 0.585 |
| 0.11 | 0.647 | 0.384 | 0.566 | **0.788** | **0.745** | 0.907 | 0.397 | 0.803 | **0.593** |
| 0.13 | 0.637 | 0.278 | 0.538 | 0.767 | 0.699 | 0.907 | 0.476 | 0.843 | 0.587 |

Table 7: Ablation study of the hyper-parameter $\tau$ of the proposed contrastive variance loss on the Synpase dataset. We report the class average HD95 results and the results for all individual classes.

| $\tau$ | HD95.Avg↓ | Aor | Gal | Kid(L) | Kid(R) | Liv | Pan | Spl | Sto |
|---|---|---|---|---|---|---|---|---|---|
| 0.05 | 84.61 | 20.74 | 79.45 | 324.51 | 88.75 | **14.33** | 22.95 | 101.75 | 24.40 |
| 0.07 | **37.38** | **15.33** | **39.26** | 53.53 | 80.30 | 28.09 | **19.61** | 40.05 | **22.88** |
| 0.09 | 84.94 | 231.11 | 49.55 | 201.54 | 78.03 | 19.28 | 21.28 | 52.11 | 26.61 |
| 0.11 | 44.44 | 31.36 | 51.40 | **38.28** | **51.13** | 33.75 | 26.81 | 99.46 | 23.30 |
| 0.13 | 64.43 | 44.24 | 145.95 | 99.64 | 103.49 | 18.23 | 44.51 | **36.06** | 23.30 |

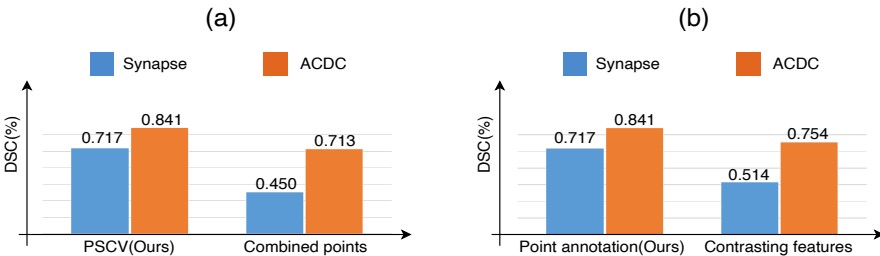

Figure 1: (a) Impact of using combined points as pseudo labels on the Synpase and ACDC datasets. (b) Impact of contrasting variance maps on the Synpase and ACDC datasets.

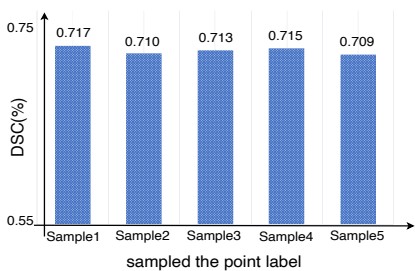

Figure 2: Robustness analysis of point-based annotation on We randomly sampled the point label five times.

This suggests that PSCV can effectively handle several random point labels and still maintain its robustness.

**Impact of the weight of the loss function.** To demonstrate the impact of weight of the loss function $\lambda_{cv}$, we summarize the results on the Synapse dataset in terms of DSC in Table 4 and HD95 in Table 5. Note that a value of $\lambda_{cv}$=0.3 can obtain the best class average DSC compared to other weights in Table 4. In this case, a low value would reduce the weight of the proposed loss while a high value would make the model focus on too many lower-level features, which is not good for the final results. At the same time, Table 5 shows that the ideal results are achieved for this HD95 for both $\lambda_{cv}$=0.07 and $\lambda_{cv}$=0.3. In order to balance both the area of the segmentation result and the accuracy of the edge points, we set $\lambda_{cv}$=0.3, which gives the desired results on both the DSC and HD95 metrics.

**Impact of the temperature of the contrastive variance loss.** To demonstrate the impact of the temperature $\tau$ of the contrastive variance loss, we summarize the results on the Synapse dataset in terms of DSC in Table 6 and HD95 in Table 7. $\tau$ is crucial for controlling the strength of penalties on hard negative samples in the contrast learning paradigm. As shown in the tables, the proposed method achieves the best results on both the DSC and HD95 metrics when we set $\tau$=0.07. In particular, Aor and Gal, as categories with smaller areas in this dataset, both achieved the best results for this parameter at the same time. And the results in the other categories have achieved satisfactory results.

## 3 ADDITIONAL VISUALIZATIONS

For better visual understanding of $z_k^n$, we have included visualizations of key variables $z_k^n$ and $c_k^n$, as shown in Figure 3. The figure portrays the evolution of $z_k^n$ for different iteration stages during model training. Notably, as depicted in the figure, $z_k^n$ progressively concentrates on the region corresponding to category k as training advances. Towards the culmination of training, $z_k^n$ effectively encapsulates the relevant category organs. Throughout the training process, our loss function facilitates the selection of positive and negative samples across images for comparative learning, ultimately realizing the intended outcome.

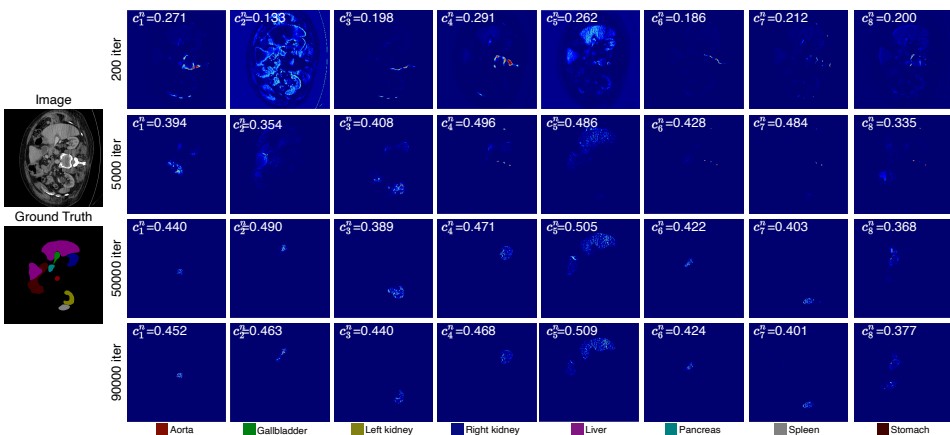

Figure 3: Visualizations of $z_k^n$ and $c_k^n$. The leftmost column represents the input image and its ground-truth. Each column in the right part represents the $z_k^n$ of the corresponding category $k$ at different training iterations. Each row denotes the $z_k^n$ of a different category for the same training iteration. The number in each subfigure indicates $c_k^n$.