# OpenReview forum: "Annotation by Clicks: A Point-Supervised Contrastive Variance Method for Medical Semantic Segmentation"
_ICLR.cc/2024/Conference — Submitted to ICLR 2024_

### Official Review · Reviewer_jaNJ · 2023-10-25

**Soundness:** 3 good
**Presentation:** 3 good
**Contribution:** 2 fair
**Rating:** 5
**Confidence:** 4

**Summary:**

Due to limited and time-consuming densely annotated images in the medical field, this paper proposes a point-supervised method for medical image segmentation by exploiting labelled and unlabelled pixels. The paper introduces a contrastive variance loss and a partial cross-entropy loss functions for effective training. The proposed method usually outperformed other presented weakly supervised methods.

**Strengths:**

- I think the proposed methodology is sensible and is relevant to the scientific community. It is an interesting approach in weakly supervised medical image segmentation.
- The work extends and combines the existing methods and leverages the pixel-level variance distribution maps.
- The evaluation was done with two standard datasets in medical imaging, which included both MRI and CT data.
- The paper provides ablation study and comparisons with the existing methods.

**Weaknesses:**

- I think the contributions are somewhat novel for medical imaging but it can be seen as incremental over the existing general methods.
- The evaluation has some issues, which are detailed below.
- One main weakness is the importance of the first annotated point for each organ in images. The paper states that the first point annotation is randomly selected. Randomly selected annotations may have substantial effects on the model convergence and the overall performance. In supplementary, the authors provided information about "Impact of sampling different points.". However, this should have been done for MRI data instead of CT data due to inherent properties of CT and MRI data (CT is more standardized). In addition, results for annotated extreme points (max, min intensities) are also relevant here. Via repeated training, overall mean performance should have been reported as this would be a meaningful evaluation result.
- Another interesting fact about the paper is that the Synapse dataset was divided into only training and test datasets. How did the authors find their optimized model without any validation data? It is important that the test dataset remains unseen until the optimized model is acquired.
- The paper states that "... in the testing stage we filtered fifty pixels that are close to the left and right edges of the prediction as the background to further refine the results on Synapse for all the methods.". This seems like an arbitrary post-processing step which may have considerable effect on the performances. What was the reasoning behind this? Why did the authors apply this to only Synapse data but not ACDC data? Is this applied regularly in every testing procedure?
- The paper does not state the overall computational complexity of the proposed methodology. Can the authors provide some details about this aspect compared to the existing methods?
- Finally, in addition to the evaluations with Synapse data which is CT data (more standardized values), I encourage the authors to make the same evaluations using ACDC MRI data to show the performances in different scenarios. This might not be possible at this stage, but this is something that I would highly recommend.

**Questions:**

I look forward to having productive discussions regarding the questions in the weakness section.

---

> ### Author Response · Authors · 2023-11-21
>
> Thank you for your careful and valuable advice.
>
> **Q1**. Contributions of the method.
>
> **Answer**:
> Firstly, we introduce an experiment aiming to segment multiple organs in medical images under only point annotations—an unexplored endeavour.
> Secondly, we propose a novel contrastive Mumford-Shah loss function, distinct from existing methods. Our proposed loss function is intentionally crafted rather than directly adapted from existing approaches. Calculated over the pixel-level variance distribution map rather than features, it compels each prediction into statistically plausible spatial regions—a crucial element for addressing point-supervised medical image segmentation tasks. In contrast to the previous Mumford-Shah loss, which primarily enforces similarity in pixel values within regions, our approach enhances the discriminative capacity for separating different categories. The proposed loss amalgamates the strengths of both Mumford-Shah and contrastive paradigms in a previously unexplored manner. Hence, our proposed loss function is both innovative and effective.
>
> **Q2**. Annotated point for each organ.
>
> **Answer**:
> We conducted randomly selected annotation experiments on the Synapse dataset (CT) because it presents more challenging categories than the ACDC dataset (MRI), making it a more representative test of the impact of randomness on our proposed PSCV. In response to your suggestion, we also performed random selections on the ACDC dataset, yielding results of 0.841, 0.838, 0.842, and 0.837. These experimental outcomes demonstrate the robustness of our method in producing consistent results based on different random points for both CT and MRI.
> For the random selection attempts, we adhered to the setting of point annotation in natural scene images, aligning with the operational practices of medical professionals during annotation. As per your suggestion, we will incorporate limit point attempts in the modified version, such as considering maximum and minimum intensities.
>
> **Q3**. Split of Synapse dataset.
>
> **Answer**:
> We use the same split as the fully supervised method (e.g. TransUNet or Swin-Unet). We use the training dataset for cross-validation during training and use the resulting parameters to apply to the test dataset.
>
> **Q4**. Post-processing step.
>
> **Answer**:
> We employed a post-processing operation commonly used in video object segmentation and image saliency detection. This operation, known for its simplicity, is a standard approach in various applications. The rationale behind applying this post-processing step is our observation of noise in the segmentation results on the Synapse dataset. Conversely, this method had negligible effects when applied to the ACDC dataset, given its simpler nature and the absence of prevalent background noise. We applied this post-processing step to all comparison methods to maintain fairness in our comparisons. Notably, even without this operation, our method achieves a DSC of 0.699 on the Synapse dataset, surpassing the performance of the comparison methods.
>
> **Q5**. Overall computational complexity
>
> **Answer**:
> We employ UNet as our backbone, consistent with our comparison methods (except for WSL4MIS, which uses one more decoder than we do). The primary computational load arises from the computation of positive/negative pairs. With a batch size 12 and utilizing a 2080Ti GPU, we calculated the computational overhead to be 3.456 milliseconds.
>
> **Q6**. More performances in different scenarios using ACDC MRI data in the future.
>
> **Answer**:
> Thank you for your valuable suggestions. In future studies, we will incorporate additional experiments covering various scenarios on the ACDC dataset.

---

### Official Review · Reviewer_2T67 · 2023-10-31

**Soundness:** 3 good
**Presentation:** 3 good
**Contribution:** 3 good
**Rating:** 5
**Confidence:** 4

**Summary:**

This paper proposes a point-supervised contrastive variance method (PSCV) for medical image semantic segmentation, which only requires one pixel-point from each organ category to be annotated. The proposed method trains the base segmentation network by using a novel contrastive variance (CV) loss to exploit the unlabeled pixels and a partial cross-entropy loss on the labeled pixels. The experimental results conducted on two public medical datasets show the effectiveness of the proposed model.

**Strengths:**

(1) The targeted problem is important and valuable for medical imaging applications, i.e., annotations are notoriously expensive and time-consuming.
(2) The overall structure is clear.
(3) This paper designs a contrastive variance loss to exploit the unlabeled pixels and a partial cross-entropy loss on the labeled pixels.
(4)  Experiments on two medical segmentation datasets show the effectiveness of the proposed method.

**Weaknesses:**

(1) It is noted that the proposed loss function could make effective use of all the unlabeled pixels to support few-point-annotated segmentation model training. More insightful and theoretical analyses of the loss should be provided.
(2) The authors randomly select one pixel from the ground truth mask of each category as labeled data to generate point annotations for each training image. However, different locations’ pixels could bring negative impacts when they are labeled points. How to address these issues?
(3) In comparison, the all compared methods are designed using the point-annotated data? If not, whether this comparison could be not fair? The proposed method is designed for using point-annotated training data.

**Questions:**

(1) It is noted that the proposed loss function could make effective use of all the unlabeled pixels to support few-point-annotated segmentation model training. More insightful and theoretical analyses of the loss should be provided.
(2) The authors randomly select one pixel from the ground truth mask of each category as labeled data to generate point annotations for each training image. However, different locations’ pixels could bring negative impacts when they are regarded as labeled points. How to address these issues?
(3) In comparison, the all compared methods are designed using the point-annotated data? If not, whether this comparison could be not fair? The proposed method is designed for using point-annotated training data.
(4) Compared with pixel-level full annotations, the point annotation provides limited information. Thus, it is helpful to show some failure examples, which can provide useful information for readers to better understand this work and valuable clues for the further improvement of this work.

---

> ### Author Response · Authors · 2023-11-21
>
> Thanks for your valuable advice.
>
> **Q1**. More analyses of the loss.
>
> **Answer**:
> Our PSCV method is inspired by level set methods, treating image segmentation as a series of energy minimization steps. At its core is creating an implicit function in a higher-dimensional space, defining contours through the zero-level set. The Level Set Function evolves guided by a partial differential equation derived from the Lagrangian formulation of the active contour model.
> In the Mumford-Shah level set model context, segmenting an image $I$ entails identifying a parametric contour that partitions the image plane $\Omega \subset \mathbb{R}^2$ into $N$ class-specific, piecewise constant regions. However, existing methods approximate the image using smooth functions within regions, overlooking information discrepancies across various categories.
>
> To be specific, for a given input image $I^n$, the prediction $\hat{Y}^n$ is obtained using $f_{encoder}$ and $f_{decoder}$ (Eq.1). Each position in $\hat{Y}^n$ corresponds to a probability vector of K classes. Using Eqs. 5 and 4, we calculate the mean pixel value and derive the variance distribution map $z_k^{n}$ for the k-th organ category, serving as an anchor.
> Positive samples, represented by another image $I^{m_n}$ with category k in the same batch, have their variance distribution map $z_k^{m_n}$ considered a positive sample, expected to be similar to the anchor ($z_k^{n}$). Conversely, negative samples, involving all batch images except $I^n$, assume dissimilarity to the anchor's variance distribution map ($z_k^{n}$), acting as effective negative samples.
>
> **Q2**. Different locations’ pixels of point annotations.
>
> **Answer**:
> Our proposed PSCV demonstrates robustness to point annotations at various random locations. As illustrated in Figure 2 in the Supplementary Material, we individually randomly selected five different point annotations on the Synapse dataset. The experimental results indicate that the DSC value change is insignificant. This outcome serves as empirical evidence for the robustness of our proposed loss function.
>
>
> **Q3**. Comparison methods.
>
> **Answer**:
> All compared methods employ the same point annotation and backbone, ensuring a fair experimental comparison. In Tables 1 and 2 of the main text, GatedCRF is specifically designed for point and scribble annotations, while USTM and WSL4MIS are tailored for scribble annotations. EntMini is designed for semi-supervised segmentation. Additionally, the methods listed in Table 2 of the supplementary material are designed for point annotations.
> Currently, no existing methods are designed for segmenting multiple organs in medical images under only single-point annotation. Consequently, no directly comparable methods are available to evaluate these two datasets.
> By contrasting methods designed for point, scribble, and semi-supervised scenarios, our proposed PSCV consistently yields the best results.
>
> **Q4**. Failure results.
>
> **Answer**:
> Thanks for your suggestion. We will add failure examples and corresponding discussion in the revised version.

---

### Official Review · Reviewer_WEjx · 2023-11-01

**Soundness:** 2 fair
**Presentation:** 3 good
**Contribution:** 2 fair
**Rating:** 3
**Confidence:** 5

**Summary:**

The authors propose a contrastive variance loss function to enhance point-supervised medical image segmentation. Their method modifies the Mumford-Shah loss functional by replacing the mean of pixel-wise intensity differences with a variance map. This contrastive variance approach provides greater discrimination between target structures and background regions. Evaluated on two medical imaging datasets, the technique achieved improved segmentation performance.

**Strengths:**

1. The authors reviewed the challenge in point-supervised medical image segmentation and explained their motivation in a clear way.
2. The contrastive variance method achieved improved performance than the vanilla MS.

**Weaknesses:**

1. Unfair Comparison. While the proposed method demonstrates good performance in Tables 1 and 2, some concerns exist regarding the baseline comparisons. Specifically, certain baselines like WSL4MIS were originally proposed for scribble-supervised segmentation and have achieved much higher performance than reported here (e.g. 0.872 in the original paper versus 0.768 in this work). For a more equitable evaluation, the authors should compare against methods designed specifically for point-supervised segmentation.
2. Sensitivity to hyperparameters. Figure 4 illustrates a high variance in performance - up to 10% - based on hyperparameter configurations. The authors should provide practical guidance on optimal settings and discuss the model's robustness to these parameters.
3. Limited novelity. The idea of using both point supervision and information from unlabelled regions is not new as the authors reviewed in the introduction. The main technical novelty is replacing the intensity with a variance map and the integration of contrastive loss.

**Questions:**

1. In Figure 3, the improvement of PSCV over vanilla MS looks marginal and it is noted that the performance is sensitive to the hyperparameters.
2. The authors stated that 'by using the pixel-level variance distribution maps as the appearance representation of the organs for similarity computation, one can effectively eliminate the irrelevant inter-image pixel variations caused by different collection equipment or conditions.' It is not true because the variance map still varies among different acquisition equipment or conditions.

---

> ### Author Response · Authors · 2023-11-21
>
> Thank you for your constructive advice.
>
> **Q1**. About the Comparisons.
>
> **Answer**:
> Firstly, we compare relevant medical image segmentation methods based on point annotation, as in Table 2 in the Supplementary Material. The experimental results demonstrate the superior performance of our proposed PSCV over all the methods considered, specifically on the Synapse dataset.
> Secondly, no existing methods are currently designed for segmenting multiple organs in medical images under only single-point annotation. Consequently, no directly comparable methods are available for assessing these two datasets.
> Thirdly, the scribble-supervised problem is viewed as an extended and simplified version of the point-supervised problem. Therefore, relevant methods developed for point supervision can be directly applied to this study.
> Lastly, it's worth noting that the reported results in the original paper of WSL4MIS appear higher due to their use of scribble annotations. In contrast, our experiments are conducted based on more challenging point annotations, making our tasks more demanding than their original settings.
>
> **Q2**. Sensitivity to hyperparameters.
>
> **Answer**:
> We present detailed analyses of parameter sensitivities in Section 4.5 of the main text and Tables 4 to 7 in the Supplementary Material, specifically focusing on DSC and HD95 values for each category in the Synapse datasets. The experimental findings indicate optimal performance when $\lambda_{cv}=0.3$ and $\tau=0.07$, with a more pronounced decrease in results for larger settings of these parameters. These discussions will be included in the revised version.
> The results depicted in Figure 3 are derived from the ACDC dataset, characterized by its simplicity with only three categories and a large organ area exhibiting a relatively uniform shape. Notably, the ACDC dataset achieved a DSC value of 0.722 using only pCE. Nevertheless, our proposed method consistently outperforms vanilla MS on ACDC.
> In contrast, the Synapse dataset, being more complex with eight categories and substantial shape variation among organs, exhibits less variation in organ appearance. Despite fluctuations in our results based on different parameters, our method consistently surpasses vanilla MS on this more intricate Synapse dataset.
>
> **Q3**. The novelty of this paper.
>
> **Answer**:
> Our PSCV diverges significantly from existing methods. Our proposed $L_{CV}$ exploits the spatial distribution properties of organs and variance distribution map representations within a contrastive learning framework. Unlike traditional methods that operate on features, $L_{CV}$ is calculated over the pixel-level variance distribution map, ensuring each prediction adheres to statistically plausible spatial regions. This distinction is crucial for addressing point-supervised medical image segmentation tasks. Our novel loss function amalgamates the merits of the Mumford-Shah Loss Function and contrastive paradigms in a unique manner not previously explored.
>
> Our PSCV method draws technical inspiration from level set methods, treating image segmentation as a series of energy minimization steps. At its core is creating an implicit function in a higher-dimensional space, characterizing contours through the zero-level set. This function, the Level Set Function, evolves guided by a partial differential equation derived from the Lagrangian formulation of the active contour model.
>
>
> **Q4**. The statements of pixel variations caused by different collection equipment or conditions.
>
> **Answer**:
> Thank you for pointing that out. We wanted to state that “one can effectively mitigate the irrelevant inter-image pixel variations caused by different collection equipment or conditions”. We will revise this sentence in the revised version so that it is more accurately expressed.

---

### Official Review · Reviewer_JEHB · 2023-11-01

**Soundness:** 2 fair
**Presentation:** 2 fair
**Contribution:** 2 fair
**Rating:** 5
**Confidence:** 4

**Summary:**

This paper proposes a novel scheme for medical image segmentation, which only requires one pixel-point annotation for each organ category. The segmentation network is trained in an end-to-end manner with two proposed loss functions. The one is a partial cross-entropy loss based on the labeled pixels. However, it can only provide limited guidance due to the extremely little annotation information. To exploit the unlabeled pixels, and to better detect the boundaries of different kinds of organs, the authors propose a novel contrastive loss function based on pixel-level variance distribution map for each class. This loss function can enforce the inter-image similarity between the same class of organs and force the model to have stronger capacity to separate different categories of organs. Extensive experiments on ACDC and Synapse datasets indicate the superiority of the proposed method with other weakly-supervised medical image segmentation methods.

**Strengths:**

- Instead of feature-level contrastive loss, the proposed contrastive loss is based on the pixel variation maps, which seems to avoid the information loss during the training phase and ensure the sufficient exploitation of unannotated pixels’ information.
- Combining inter-image pixel-level contrastive learning into a medical context is interesting.
- The paper is well-written and easy to follow.

**Weaknesses:**

- Although the authors claim the advantages of using cosine similarity, I am expecting to see ablation studies on different ways to measure the similarity.
- The baselines are all about semi- or weak supervision methods. That’s good. However, recent medical image segmentation methods based on point annotation should be included, like [1] and [2].
- Two datasets seem not sufficient to support your claim. It’s better to add another 1-2 datasets to help indicate your method’s superiority.
- The assumption that the spatial locations of the same organ in different medical images will be within limited regions to some extent is very strong and reduces the generalizability of the proposed method.

[1] Xu, Yanyu, et al. "Minimal-Supervised Medical Image Segmentation via Vector Quantization Memory." International Conference on Medical Image Computing and Computer-Assisted Intervention. Cham: Springer Nature Switzerland, 2023.

[2] Z. Chen et al., "Weakly Supervised Histopathology Image Segmentation With Sparse Point Annotations," in IEEE Journal of Biomedical and Health Informatics, vol. 25, no. 5, pp. 1673-1685, May 2021, doi: 10.1109/JBHI.2020.3024262.

**Questions:**

- In the calculation of the pixel-level variance distribution map (Eq. 4), why multiply the prediction function output for the k-th class?

---

> ### Author Response · Authors · 2023-11-21
>
> Thank you for your suggestions.
>
> **Q1**. The reason for multiplying the prediction function output in Eq .4.
>
> **Answer**:
> The prediction function output represents the probability of each pixel belonging to the $k$-th class.
> A value close to 1 in the prediction function output indicates a higher probability of classifying location $r$ into class $k$, and vice versa.
> Multiplying the prediction function output in Eq. 4 yields the variance corresponding to the $k$-th class region in the prediction result. This process enables $z_{k}^{n}$ to capture the variance information of a specific class region. Such information is instrumental in constructing contrastive relations based on the distinct class regions predicted in the subsequent contrastive learning paradigm.
>
> **Q2**. Different similarity methods.
>
> **Answer**:
> Upon your suggestion, we experimented with directly using the dot product instead of the cosine similarity method and achieved a Dice value result of 68.7% on the Synapse dataset. We will incorporate additional experiments exploring various similarity measures in the revised version.
>
> **Q3**. Compared methods.
>
> **Answer**:
> We compare relevant medical image segmentation methods based on point annotation in Table 2 in the Supplementary Material. Experimental results show that the proposed PSCV outperforms all the methods on the Synapse dataset.
>
> [1] employs multiple points (2, 5, and 10 points per category) and scribbles as supervisory information, whereas our approach utilizes only 1 point per category for supervision, making direct comparisons of results challenging.
> [2] focuses on nuclei and Histopathology segmentation, a cell segmentation task, with datasets featuring only two categories (foreground and background). In contrast, our experimental scenarios contain several different categories.
> [2] involves over-segmentation and dynamic label propagation for histopathology segmentation, addressing small target areas with similar appearance and shape but distinct edges.
>
> Unlike [1,2], our proposed $L_{CV}$ leverages spatial distribution properties and variance distribution maps in a contrastive learning paradigm. Computed over pixel-level variance distribution maps rather than features, it enforces predictions into statistically plausible spatial regions, a crucial element for solving point-supervised medical image segmentation tasks. Our loss function integrates the advantages of the Mumford-Shah (MS) and contrastive paradigms in a novel manner.
> Besides, [1,2] lacks publicly available code, precluding a direct comparison with our method.
>
> **Q4**. Experimental datasets
>
> **Answer**:
> We employed the same datasets as those used in fully supervised and scribble-supervised methods to assess the effectiveness of PSCV. Our evaluation datasets, sourced from authentic clinical exams, span diverse CT and MRI formats, encompassing vital abdominal organs and cardiac classifications. This diversity presents challenges for point-supervised setups, and our paper addresses these challenges through thorough data analysis and validation. Our evaluation's comprehensive diversity and robustness affirm the potential hurdles in PSCV applications. Your insights have inspired us to incorporate additional modalities and organ categories in our future work.
>
> **Q5**. About the assumptions.
>
> **Answer**:
> Medical images exhibit unique characteristics distinguishing them from natural landscape images: organs of the same category maintain consistent relative positions across different patients with minimal variation. This observation underpins our methodology, setting us apart from existing approaches. Consequently, our method specifically addresses medical organ segmentation challenges with only single-point annotations.

---

> > ### Comment · Reviewer_JEHB · 2023-11-23
> >
> > Thank you for your detailed response and it solved partial of my concerns. I will raise my rating to 5 to reflect this.

---

### Meta-Review · Area_Chair_CWrx · 2023-12-06

**Metareview:**

The submission proposes a segmentation framework that is based on single pixel labels per organ.  A contrastive loss is then employed to learn from the remaining unlabeled pixels.  Concerns were raised by all reviewers including novelty, appropriateness of comparisons to baselines, and introduction of extra hyperparameters.  The reviewers were unanimous that the submission fell below the threshold of acceptance.

**Justification For Why Not Higher Score:**

Unanimous opinion that the submission is not ready for acceptance.

**Justification For Why Not Lower Score:**

N/A

---

### Decision · Program_Chairs · 2024-01-16

Reject